# Synergy, Additivity and Antagonism between Esculetin and Six Commonly Used Chemotherapeutics in Various Malignant Melanoma Cell Lines—An Isobolographic Analysis

**DOI:** 10.3390/molecules28093889

**Published:** 2023-05-05

**Authors:** Paula Wróblewska-Łuczka, Agnieszka Góralczyk, Jarogniew J. Łuszczki

**Affiliations:** Department of Occupational Medicine, Medical University of Lublin, ul. Jaczewskiego 8b, 20-090 Lublin, Poland; paula.wroblewska-luczka@umlub.pl (P.W.-Ł.); agnieszka.goralczyk@umlub.pl (A.G.)

**Keywords:** malignant melanoma, coumarin, esculetin, isobolographic analysis

## Abstract

(1) Malignant melanomas are dangerous skin cancers, and the treatment of melanomas with various cytostatic drugs often causes side effects and after their prolonged use resistance to these drugs appears. The aim of this study was to evaluate the anticancer effects of esculetin (a simple coumarin) and to assess pharmacodynamic interactions between esculetin and six commonly used cytostatic drugs (cisplatin, epirubicin, docetaxel, paclitaxel, mitoxantrone and vemurafenib) using an isobolographic analysis. (2) The experiments were carried out on four human malignant melanoma cell lines (FM55P, A375, FM55M2 and SK-MEL28). The effects of esculetin on viability, cell proliferation and cytotoxicity were verified in the range of concentrations of 2–200 μM. (3) Esculetin inhibited, in a dose-dependent manner, malignant melanoma cell viability and proliferation. The IC_50_ for esculetin ranged from 18.20 ± 2.93 to 120.64 ± 30.39 μM depending on the melanoma cell lines used. The combinations of esculetin with epirubicin and vemurafenib showed antagonistic interactions, the combinations of esculetin with cisplatin, docetaxel and paclitaxel showed additive interactions. For the combinations of esculetin with mitoxantrone, the isobolographic analysis displayed synergy. (4) In the treatment of malignant melanoma, esculetin should not be combined with epirubicin or vemurafenib, due to the reduction of their anticancer effects, while the synergistic interactions (esculetin + mitoxantrone) deserve a preclinical recommendation as a beneficial combination during anticancer therapy.

## 1. Introduction

Malignant melanomas are cancers that arise from melanocyte cells. Among skin cancers, they cause the most deaths of patients and, due to the distribution of melanocytes to various tissues, they can occur almost anywhere on the body, which also proves the high metastasis ratio of this cancer. For the fair-skinned Caucasian population, the origin of cancer is associated with exposure to sunlight and its ultraviolet (UV) radiation [1]. Estimates for 2020 for 185 countries of the world showed the number of new cases of melanoma at a level of 324,635 cases, while deaths amounted to 57,043 people. Skin melanoma accounts for 1.7% of all diagnosed cancers, ranking nineteenth in terms of frequency of diagnosis [2]. Surgical excision is the main method of treatment, and in the case of metastasis or resistance, other methods of treatment are available. Understanding the mechanisms and cellular pathways allow for the development of more effective therapies. These are: immune checkpoint inhibitors, targeted therapies based on signaling pathways, intrafocal therapy (for irremovable melanomas) and local therapy (for single melanomas) [3]. A new trend of research concerns the testing of plant-derived compounds as a complementary therapy or as an adjuvant treatment against melanomas. These substances contribute to the modulation of biochemical pathways and processes that induce carcinogenesis. They have the ability to elicit many types of biological responses. They strengthen the reaction of the immune system and protect against free radical damage to living cells. They can be used not only in the prevention of cancer but also in its treatment. Tested individually (for a better understanding of the mechanisms of anticancer activity) or in combination with other substances, they become an interesting target for the development of more beneficial forms of therapy [4,5,6].

An interesting compound of natural origin, commonly used as a herbal medicine in Asian countries is esculetin (aesculetin) [7]. Esculetin is a simple coumarin with two hydroxyl groups at carbons 6 and 7; it is also an aglycone metabolite of esculin (Figure 1).

Esculetin is present in many plants, such as *Citrus limon*, *Euphorbia lathyrism*, *Ceratostigma willmottianum*, *Aesculus hippocastanum* [8], and *Fraxinus rhynchophylla*—extracts from these plants are used by herbalists as traditional astringents and medicines for liver and gallbladder ailments [9]. Esculetin is characterized by many biological properties, including strong antioxidant (ability to capture reactive oxygen species—ROS) and anti-inflammatory properties. Esculetin is considered a potential drug candidate in the treatment of diabetes and its complications, atherosclerosis, atopic dermatitis, and as an anticoagulant [9,10,11]. Esculetin exhibits dual regulation of apoptosis, including anti-apoptotic activity (associated with antioxidant and anti-inflammatory effects) [8,9], and inducing apoptosis, inhibiting the proliferation of cancer cells, which is associated with the great potential of this compound as an anticancer drug [12,13].

Both in vivo and in vitro studies confirm that the mechanism of induction of apoptosis of cancer cells by esculetin is associated with an increased percentage of DNA fragmentation and arrest of the cell cycle in the G1 phase. This is related to the upregulation of p21WAF1 and p27KIP1 expression, leading to a decrease in cyclin D1/cyclin-dependent kinase (CDK) 4 and cyclin E/CDK2 complexes [14,15].

## 2. Results

### 2.1. The MTT Assay

In the MTT test, it was observed that all the tested drugs (i.e., esculetin, cisplatin, epirubicin, docetaxel, paclitaxel, mitoxantrone and vemurafenib) inhibited the viability of four malignant melanoma cell lines (Figure 2a–d). Additionally, the effect of esculetin on the viability of human normal melanocytes and keratinocytes was reported (Figure 2e,f).

Significant inhibition of malignant melanoma cell viability was observed already from the concentrations of 2–10 µM esculetin, and at higher concentrations of esculetin up to 200 µM, viability was observed reaching 20–25% for the FM55P and FM55M2 cell lines and about 30–40% for the A375 and SK-MEL28 cell lines. Esculetin had a slight effect on the viability of normal keratinocyte and melanocyte cell lines, a significant inhibition of viability was observed above concentrations of 60–100 µM. The experimentally derived median inhibitory concentration (IC_50_) values for esculetin, cisplatin, epirubicin, docetaxel, paclitaxel, mitoxantrone and vemurafenib in various melanoma cell lines are presented in Table 1. Of note, the IC_50_ values for cisplatin, mitoxantrone, docetaxel and paclitaxel were determined experimentally in our previous studies [5,16,17].

### 2.2. The BrdU Assay

Depending on dose, esculetin administered alone reduced the proliferation of FM55P, A375, FM55M2 and SK-MEL28 malignant melanoma cell lines and normal human melanocytes (HEMa-LP) and keratinocytes (HaCaT). The treatment of all the investigated cells with increasing concentrations of esculetin reduced DNA synthesis that was evaluated by measuring BrdU (5-bromo-2′-deoxyuridine) incorporation into cellular DNA in proliferating cells (Figure 3).

Esculetin from the concentration of 10–20 µM inhibited significantly malignant melanoma cell proliferation. In a concentration of 100 µM and higher, esculetin inhibited almost 100% of melanoma cell proliferation. In the case of normal melanocyte and keratinocyte lines, inhibition of proliferation of cell lines was also observed, but much less compared to malignant melanoma cell lines. In the case of the HEMa-LP melanocyte line, esculetin (at a concentration of 150–200 µM) significantly reduced cell proliferation to approximately 50–40%. In the case of the HaCaT keratinocyte line, a significant inhibition of proliferation by esculetin was observed from a concentration of 40 µM, however, a reduction of proliferation below 50% was observed only for esculetin at a concentration of 200 µM (Figure 3).

### 2.3. The LDH Assay

Cytotoxicity of esculetin to normal human melanocytes (HEMa-LP), normal human keratinocytes (HaCaT) and four malignant melanoma cell lines (FM55P, A375, FM55M2, SK-MEL 28) was detected by means of LDH assay. The release of lactate dehydrogenase into the medium, as a consequence of cell membrane destruction and cell death after exposure to esculetin, was estimated in the LDH assay [18]. In this test, esculetin in the concentration range of 20–200 µM produced a significant LDH leakage in the FM55P and SK-MEL28 malignant melanoma cell lines (Figure 4a,b). Additionally, esculetin in the concentration range of 40–200 µM for the cell line A375, and 100–200 µM for the cell line FM55M2, produced a significant LDH leakage, respectively (Figure 4c,d). In the case of normal human cells, esculetin turned out to be cytotoxic at high concentrations above 150 µM for keratinocytes and 200 µM for melanocytes (Figure 4e,f).

### 2.4. Pharmacological Interactions between Esculetin and Cisplatin, Epirubicin, Docetaxel, Paclitaxel, Mitoxantrone and Vemurafenib

From the linear log-probit concentration–response inhibitory effects for esculetin and six commonly used chemotherapeutics on four malignant melanoma cell lines, it was possible to determine the median inhibitory concentrations (IC_50_ values ± SEM). In such a situation, the drug concentrations were transformed to logarithms and their anti-proliferative effects (measured by the MTT test) were transformed to probits. The linearly related concentration–response effects for the studied drugs (when used alone and in combinations) were plotted into the Cartesian system of coordinates (Appendix A). The experimentally derived median inhibitory concentration (IC_50_) values for tested compounds are presented in Table 1.

The next step was to perform a test for parallelism of the experimentally determined concentration–response effect lines for all the tested combinations of esculetin with one of the following drugs: cisplatin, epirubicin, docetaxel, paclitaxel, mitoxantrone, vemurafenib (Appendix A). The log-probit analysis revealed that esculetin had its concentration–response line parallel to cisplatin and epirubicin in all the tested cell lines (Appendix A). In the case of the combination of esculetin and docetaxel, the concentration–response lines are parallel only for the cell line A375 (Appendix A), and in the case of the combination of esculetin with paclitaxel, they are parallel for the cell lines FM55P, A375 and SK-MEL28 (Appendix A). For the remaining cell lines, the combinations of esculetin with docetaxel and paclitaxel show non-parallel concentration–response lines (Appendix A). In the case of the combination of esculetin with mitoxantrone, the concentration–response lines are parallel for all the investigated cell lines (Appendix A), and in the case of the combination of esculetin with vemurafenib, they are parallel for the A375 and SK-MEL28 cell lines (Appendix A). Whether the concentration–response lines are parallel or not affects the appearance of the additivity line on the isobologram.

Isobolographic analysis of interactions for parallel concentration–response lines revealed that the combination of esculetin with cisplatin (at the fixed ratio of 1:1) exerted additive interactions in all the studied melanoma cell lines (Figure 5a,c,e,g). The IC_50mix_ value for this combination did not differ from the IC_50add_ value (Student’s *t*-test with Welch’s correction). This is also due to the close location of points A (IC_50add_) and M (IC_50mix_) on the isobologram. The combination of esculetin and epirubicin (at the fixed ratio of 1:1) showed a statistically significant antagonistic interaction in the FM55P, A375, FM55M2 cell lines (Figure 5b,d,f) and additivity with a tendency towards antagonism in the SK-MEL28 cell line (Figure 5h). This is due to the comparison of IC_50mix_ values with their respective IC_50add_ values (Student’s *t*-test with Welch correction). Graphically, Point M was placed drastically above the line of additivity and Point A.

The combinations of esculetin and docetaxel (Figure 6a,c,e,g) and esculetin with paclitaxel (Figure 6b,d,f,h), both at a fixed ratio of 1:1, showed additive interactions for every tested melanoma cell line. This was confirmed by a statistical comparison of the IC_50mix_ value (point M on the isobologram) with IC_50add_ value (point A—in the case of parallel lines, or points A′ and A″ in the case of non-parallel lines) with Student’s *t*-test with Welch’s correction. On the isobolograms, the proximity of point M to point A or A′/A″ indicates an additive interaction.

With isobolography, a synergistic interaction was observed between esculetin and mitoxantrone (at a fixed ratio of 1:1) in the A375, FM55M2 and SK-MEL28 melanoma cell lines (Figure 7c,e,g). The combination of esculetin with mitoxantrone produced an additive interaction with a tendency towards synergy in the FM55P cell line (Figure 7a). The synergy on the isobologram is observed when Point M is placed drastically below the additivity line and Point A, which was also statistically confirmed with a Student’s *t*-test with Welch’s correction. On the contrary, the antagonistic interaction was observed between esculetin and vemurafenib (at a fixed ratio of 1:1) in all the tested melanoma cell lines (Figure 7b,d,f,h). As already mentioned, antagonism on the isobologram was observed when Point M is located drastically above the additivity line and Point A (for parallel lines) or the upper isobole of additivity (A″), in the case of non-parallel lines, which was also confirmed by statistical analysis.

Additionally, to strengthen the confirmation of observed interactions in this study and to provide more reliable proof of synergistic and antagonistic interactions, the interaction index (as a ratio of IC_50mix_ and IC_50add_) values for all the studied combinations of esculetin with six chemotherapeutics (epirubicin, vemurafenib, mitoxantrone, cisplatin, paclitaxel and docetaxel) were calculated. The interaction indices for antagonistic interactions between esculetin and epirubicin ranged from 1.75 to 2.16, while those between esculetin and vemurafenib ranged from 2.11 to 2.13 (Table 2). The synergistic interactions observed for the combinations of esculetin with mitoxantrone have interaction index values ranging from 0.55 to 0.27 (Table 2). The remaining additive interactions between esculetin and cisplatin, docetaxel and paclitaxel had indices ranging from 0.92 to 1.33 (Table 2).

To illustrate and summarize the interactions for mixtures of esculetin with one of the six cytostatic drugs we used a graph (in a shape of polygonogram) which represents a two-drug combination (Figure 8). With such a modified polygonogram, we found that the most favorable combinations between esculetin and six commonly used chemotherapeutics were for the mixtures of esculetin with mitoxantrone which offered synergistic interaction in three cell lines (A375, FM55M2 and SK-MEL28) and a tendency to synergy in the FM55P cell line (Figure 8).

## 3. Discussion

The results of this study confirmed that esculetin inhibits the proliferation of malignant melanoma cell lines. Many studies confirm the anticancer properties of esculetin. For instance, esculetin at concentrations of 9.3–600 μM inhibited the proliferation of several prostate cancer cell lines (PCa PC3, DU145 and LNCaP) [19]. The molecular mechanism of action of esculetin is based on induction of apoptosis, arrest of the cell cycle in the G1 phase, induction of cytochrome c, p53, p21 and p27 expression and reduction of CDK2 and CDK4 expression [19]. Esculetin is also active against pancreatic cancer. It was tested on three different cell lines of this tumor (PANC-1, MIA PaCa-2 and AsPC-1), where the effect of this compound on the induction of apoptosis (depending on mitochondria through the activation of caspases 3, 8 and 9) and cell cycle arrest was also demonstrated in Phase G1. Esculetin impedes the binding interaction between Nrf2 and KEAP-1 by activating the antioxidant response element (ARE) pathway and attenuating NF-κB activity leading to apoptosis. For pancreatic cancer cells, the IC_50_ for esculetin was 100 μM [20], which is comparable to the obtained IC_50_ values for esculetin against malignant melanoma cells ranging from 18.20 μM to 120.64 μM depending on the cell lines. A much higher concentration of IC_50_ = 400 μM (after 48 h incubation) was obtained by scientists testing esculetin on colon cancer LoVo cell line [21]. The mechanism of the action of esculetin was associated with the induction of apoptosis (increase in the amount of pro-apoptotic proteins: Bax, cleaved caspase -3, -7, -9 and poly(ADP-ribose) polymerase; decrease in the amount of anti-apoptotic proteins (Bcl-2)) and cell cycle arrest in phase G0/G1. Increased amounts of cell cycle inhibitory proteins (p53, p27 and p21) and a decrease in the amount of the progressive protein cyclin D1 were observed [21]. Another experiment also confirmed the effectiveness of esculetin against colon cancer cells, causing the activation of MAPK, caspase-3 and 9 signaling pathways, leading to apoptosis [22]. Low IC_50_ concentrations of esculetin in the range of 1.96–12.88 µM were observed against laryngeal carcinoma cells (Hep-2, TU-212 and M4e) [23]. Esculetin has been shown to significantly inhibit STAT3 phosphorylation and block STAT3 translocation into the nucleus. Esculetin also blocks the cell cycle in the G1/S phase. In addition, in vivo tests have proven that esculetin reduced the growth and weight of laryngeal cancer xenograft tumors [23].

Esculetin at concentrations up to 80 µM reduced the proliferation of Lewis lung carcinoma (LLC) cells and reduced the expression of c-myc, cyclin D1 and NF-κB [24]. After cells were injected into mice, tumor size and weight were reduced in esculetin-treated mice compared with untreated mice [24]. The IC_50_ concentrations of esculetin against melanoma (18.20–120.64 μM) obtained herein are comparable to the IC_50_ concentrations of 95–142.5 μM obtained against endometrial cancer cells (HEC-1B and Ishikawa cell lines). Esculetin promoted cell apoptosis by upregulating expression levels of cleaved caspase 3 and cleaved PARP [25].

Esculetin (at a concentration of 50 mM) was effective against HepG2 liver cancer cells, by binding to three potential targets (phosphoglycerate kinase 2 (PGK2), glycerol-3-phosphate dehydrogenase (GPD2) and glucose-6-phosphate isomerase (GPI)). Esculetin significantly inhibited the rate of glycolysis, which affects the production of lactate and glucose utilization in HepG2 liver cancer cells. Upregulation of caspase-3 and Bax, and downregulation of Bcl-2 were also observed, indicating that esculetin induced cell apoptosis. Animal tests (BALB/c mice, treated with esculetin for 15 days at a daily dose of 25 and 50 mg/kg body weight) showed that esculetin inhibited tumor growth compared to the control group. Measurement of the target protein in tumor tissue showed an increase in PGK2, GPD2 and GPI in response to esculetin treatment [26]. Esculetin at a concentration of 20 µM inhibited the proliferation of leukemic cells and induced autophagy through the formation of autophagic vesicles. In addition, a decrease in the expression of cyclins D1, D3, DK4 and DK2 was observed, resulting in cell cycle arrest in the G0/G1 phase. Esculetin has been shown to block the phosphorylation of MAPK and extracellular signal-regulated kinase (ERK), thereby inhibiting the activation of Raf/MAPK/ERK signaling [27]. Esculetin (at a concentration of 850 µM) inhibited the proliferation of MGC-803 gastric cancer cells by reducing the potential of the mitochondrial membrane, while activating the mitochondrial apoptotic pathway [28]. Inhibition of the IGF-1/PI3K/Akt pathway by esculetin resulted in the activation of caspase-3 and -9, as well as an increase in the Bax/Bcl-2 index and the release of cytochrome c from mitochondria [28]. Another experiment confirmed that esculetin raises the level of calcium in the cytoplasm by releasing it from the endoplasmic reticulum (ER), thereby activating the calcium-related mitochondrial apoptosis pathway that promotes apoptosis of human cancer cells [29,30]. Esculetin is also effective against HeLa cervical cancer cells (IC_50_ = 37.8 µM) by inducing their apoptosis, as evidenced by the production of reactive oxygen species (ROS) and the accumulation of cells in the sub-G1 phase [31]. Esculetin inhibits the proliferation of oral squamous cell carcinoma cell lines (HN22 and HSC4) when tested at concentrations of 5–20 µg/mL. The cell cycle arrest in the G1 phase and apoptosis were observed, as a result of suppression of the transcription factor Sp1 (Sp1) and modulation of genes p27 and p21, and cyclin D1 in response to esculetin treatment [32].

It is interesting to note that esculetin shows strong inhibition (IC_50_ = 43 µM) of the key enzyme of melanogenesis, which is the fungal tyrosinase. In addition, esculetin at a relatively low concentration of 5 µM (compared to concentrations tested in our experiment in the range of 2–200 µM) significantly inhibited melanin production in B16 murine melanoma cells without affecting their growth. In addition, it was proved in experiments on patches of split epidermis, which were treated with esculetin solutions, that the number of melanocytes decreased in them (compared to the control), which additionally confirmed the inhibitory effect on tyrosinase activity and the process of melanogenesis [33]. Jeon et al. confirmed the inhibition of proliferation of the human melanoma G361 cell line (IC_50_ = 42.86 μg/mL ≈ 240 μM), which exceeds the IC_50_ concentrations we obtained for four melanoma cell lines. However, scientists proved that esculetin induced nuclear condensation and fragmentation, induced apoptosis by reducing the level of Sp1 protein and stopped the cell cycle. Esculetin increased protein levels of p21, p27, Bax and active caspase 3 and decreased protein levels of cyclin D1, procaspase 3 and PARP [34]. In in vitro experiments on B16F10 mouse melanoma cells, esculetin at a concentration of 50 µM was observed to be an inhibitor of the cyclooxygenase and lipoxygenase pathways (related to arachidonic acid metabolism), rendering the cells non-invasive by reducing the production of matrix metalloproteinase 2 (MMP-2), an enzyme required for membrane degradation. This fact was confirmed in a mouse model; the administration of esculetin reduced the formation of tumor metastases to the lungs compared to the control [35].

As mentioned in the Introduction, esculetin, in addition to its pro-apoptotic effects on cancer cells, also has an apoptosis-inhibiting effect. A protective effect against oxidative stress was observed in mouse C2C12 muscle cells and rat H9c2 cardiomyocytes [36,37]. Esculetin (at concentrations of 10–100 µM) protected against oxidative stress, increased cell viability, and reduced apoptosis of human HCE corneal epithelial cells compared to controls. Esculetin regulated the expression of Bcl-2, Bax and caspase-3 proteins, and strongly induced the translocation of Nrf-2 to the nucleus, which in turn enhanced the antioxidant genes regulated by Nrf2 signaling (HO-1, NQO1, GCLM, SOD1 and SOD2) in HCE cells treated with H_2_O_2_ [38]. Our experiments also confirmed that esculetin in concentrations up to 100 µM did not significantly reduce viability and was not cytotoxic to normal cells such as human HaCaT keratinocytes and HEMa-LP melanocytes.

Generally, the first line of malignant melanoma therapy is associated with surgical excisions. In the case of metastases, recurrence or inoperable melanoma changes, either radiotherapy or chemotherapy methods are used. Most of the cytostatic drugs used are characterized by numerous side effects or drug resistance that appears after use for some time. For example, vemurafenib (a BRAF inhibitor) is an effective cytostatic drug used in the treatment of melanoma (due to frequent BRAF V600E mutations, detected in about 40% of patients), however, resistance is observed after 6–10 months of therapy, and the therapy itself is associated with side effects, including joint pain, headache, fatigue and fever. There are also alarming observations that some patients develop other skin cancers after monotherapy: squamous cell carcinoma or keratoacanthoma [39,40]. For these reasons, multidrug therapy of malignant melanoma is being considered more often. Due to the limited side effects of substances of natural origin, combinations of natural compounds with cytostatic drugs are tested pre-clinically. In oncology, the general principle is based on a combination of drugs that produce their anticancer effects by different molecular mechanisms of action, so that the drugs can work together to inhibit malignant melanoma cell proliferation. From a pharmacological point of view, three main types of interactions between anticancer drugs can be distinguished: supra-additivity (synergy), additivity and subadditivity (antagonism) [5,17,41,42].

In the present study, the interactions between esculetin (a naturally occurring simple coumarin) and several cytostatic drugs with different mechanisms of action were evaluated. Since the cell lines we tested have the BRAF mutation, vemurafenib was included in the tested combinations. This compound has the ability to inhibit mutated BRAF serine-threonine kinase and selectively binds to the ATP binding site of BRAF V600E kinase and inhibits its activity. Vemurafenib potently inhibits ERK phosphorylation and proliferation of BRAF mutant cells only [40]. The study also included taxanes—very effective cytostatic drugs belonging to the group of mitotic poisons, whose mechanism of action is related to the stabilization of microtubules. Docetaxel is a stronger derivative of paclitaxel than the maternal drug and is more water soluble [16,43]. The mechanism of action of cisplatin is the formation of covalent adducts between platinum complexes and DNA, but also with RNA and many proteins, which leads to the induction of apoptosis [44]. Mitoxantrone intercalates in cellular DNA, causes DNA strand breakage and inhibits DNA synthesis and transcription, leading to cell apoptosis [45]. Epirubicin also interferes with DNA synthesis and function, but is most active in the S phase of the cell cycle [46].

Thus, our experiments and isobolographic analysis showed that the combinations of esculetin and cisplatin are characterized by additive interactions. An additive interaction was also observed for the combination of cisplatin and osthole for two melanoma cell lines, FM55P and FM55M2 [5], and for the combination of cisplatin with fraxetin (7,8-dihydroxy-6-methoxycoumarin) for four malignant melanoma cell lines [6]. Our experiments showed that the combinations of esculetin with taxanes (paclitaxel and docetaxel) also showed additive interactions, although the combination of fraxetin with docetaxel showed an additive interaction with a tendency to antagonism [6], which may be due to the different chemical structure of fraxetin compared to esculetin. Antagonism is an undesirable interaction in therapy from a pharmacological point of view; one drug inactivates the other in the combination, thereby weakening their anticancer effects [47,48]. For the combination of esculetin with epirubicin and vemurafenib, statistically significant antagonistic interactions were observed for all tested melanoma cell lines, which allows us to conclude that these compounds should not be combined. Synergism is the most desirable pharmacodynamic interaction as the combined compounds potentiate their anticancer effect, which means that lower doses of the compounds can be used to achieve a specific therapeutic effect [47,48]. Esculetin, in combination with mitoxantrone, showed a synergistic interaction against malignant melanoma, which deserves a preclinical recommendation as a very beneficial combination during anticancer therapies.

There is no doubt that the isobolographic analysis of interaction is a method of choice for characterizing the interactions between drugs or agents whose molecular mechanisms of action are unknown or not precisely identified as yet [49,50,51,52]. Although three types of interactions (i.e., synergy, additivity and antagonism) are clearly defined in isobolography, it is difficult to assess the strength of such interactions [50]. The statistical analysis only provides information whether or not the experimentally derived IC_50mix_ values considerably differed from the theoretically calculated IC_50add_ (accepted as additive) values. Theoretically, it is possible to statistically assess and differentiate the significance between the analyzed IC_50_ values (i.e., *p* < 0.05 vs. *p* < 0.0001), but this differentiation is not precise. For instance, a synergistic interaction assessed at *p* < 0.0001 is more powerful than that at *p* < 0.05. Analogously, an antagonism at *p* < 0.05 is less powerful than that observed at *p* < 0.0001. Of note, the interaction indices were calculated in this study to determine the power (strength) of interactions [48]. Generally, it is observed that the lower the interaction index, the greater the synergy. In the case of antagonism, the higher the interaction index, the more powerful the antagonism is observed for the studied combination [48,50,51]. However, in our study, the arbitrary accepted border values of the interaction indices for synergy (0.7) and for antagonism (1.3) were not precisely adjusted to our results. The interaction index value of 0.55 was calculated for the combination of esculetin with mitoxantrone for the FM55P cell line, albeit the IC_50_ values did not differ significantly (Table 2). Additionally, the interaction index amounting to 1.75 was calculated for the combination of esculetin with epirubicin for the SK-MEL 28 cell line, which was associated with no statistical significance (Table 2). According to the arbitrary accepted border values of interaction indices (0.7 and 1.3), the mentioned combinations should indicate synergy and antagonism, respectively, but statistical analysis revealed no significance. Although the statistical analysis of data proved insignificant, the calculated interaction indices were considerably out of the accepted border range for synergy (0.7) and antagonism (1.3), thus we applied the term “tendency” towards synergy and antagonism to pay attention to this problem. This was the main reason to combine both methods together, i.e., statistical analysis of data in isobolographic analysis of interactions and calculation of interaction indices so as to strengthen the determination of the exact types of interactions in this study. It seems that both statistical analyses of data accompanied by calculation of the interaction index for the studied combinations can mutually complement both methods and should be recommended in further studies investigating the interactions among chemotherapeutics.

Theoretically, the synergistic interactions between esculetin and mitoxantrone can be readily explained in relation to a mutual potentiation of their molecular mechanisms of action. It seems that esculetin and mitoxantrone concomitantly complement one another in terms of suppression of cell proliferation and induction of apoptosis in all the tested malignant melanoma cell lines in the in vitro MTT assay. In contrast, esculetin with vemurafenib or epirubicin interacted antagonistically, which means that more concentrations of the drugs are required to assess the same anti-proliferative effects as for the drugs when used alone. It is highly likely that esculetin and vemurafenib can mutually compete on the same target sites or one drug can be replaced by another and thus antagonistic interactions occur. In such a situation, some parts of the drugs become inactive and more concentrations are needed to yield the same anticancer effects in the in vitro MTT assay. Another explanation is also possible when considering antagonistic interaction between esculetin and epirubicin with respect to their anticancer activity. It is highly likely that one drug can inhibit or slow down certain cellular processes in the cell cycle in such a manner that the second drug is unable to produce its anticancer effects because the cells are not in the right phase of the cellular cycle in which the second drug could act. As mentioned earlier, esculetin is able to arrest the cell cycle in the G1 phase, making epirubicin unable to affect its target site and interfere with DNA synthesis in the S phase of the cell cycle. This two-drug combination is a good example of the antagonistic interaction. Although such theoretical considerations are speculative, they can readily explain the observed antagonistic interactions between chemotherapeutics, but such explanations should always be experimentally verified in molecular studies to definitively confirm them.

Noteworthy, in this in vitro study, all the combinations were tested at the fixed-ratio concentration of 1:1. In isobolography, the drug combinations are very often preferentially chosen at a ratio of 1:1 as the first anticancer screening for drugs producing equally effective anti-proliferative effects. Sometimes many more combinations are required to determine the exact nature of interactions between drugs, but generally, the fixed ratio of 1:1 is always tested. In clinical conditions, various drug ratios are used based on doses of particular drugs given to the patients. For instance, if one drug is administered in a dose of 30 mg and the second in a dose of 5 mg, the ratio combination is 30:5 or equivalently 6:1. In contrast, in isobolographic studies, the ratio combination is constant in the mixture and thus the concentrations of both drugs increase along with the observed increase in the anti-proliferative effects of the drugs. It means that concentrations of the drugs in the mixture increase proportionally, but the drug ratio does not change. Previously, we have reported in some isobolographic studies that dealt with the anticonvulsant effects of the antiseizure medications that the same two-drug combination can exert synergistic, additive and even antagonistic interactions, depending on the fixed-ratio combinations tested [53]. However, in the anticancer isobolographic studies, it is important to detect drug-drug combinations producing favorable anti-proliferative effects. Researchers preferentially select a fixed ratio of 1:1 in their studies so as to detect whether or not both drugs can cooperate in terms of suppression of proliferation and/or induction of apoptosis. The combination at the fixed ratio of 1:1 provides us with information that both drugs exert their effects in the equivalent concentrations. In opposition to this could be, for instance, a combination with a fixed ratio of 1:5, where one drug could not exert its anti-proliferative effects because of its too low concentration being under the anticancer effect level. This was the main reason to test drug combinations at a fixed ratio of 1:1 as a first screening in in vitro MTT assay.

Of note, the two-drug combinations with the antagonistic interactions observed in this study are unfavorable and can be recommended neither to further preclinical experiments, nor clinical studies. Additionally, a special warning should be addressed to researchers, clinicians and potential patients to not use those combinations which exert antagonistic interactions because of the high risk of the occurrence of adverse side effects related to high concentrations of the drugs used in combinations. In contrast, the two-drug combinations offering synergistic interactions observed in this study deserve recommendation as favorable combinations, worthy of being tested in further preclinical studies and clinical conditions.

## 4. Materials and Methods

### 4.1. Cell Culture

All in vitro experiments were conducted on certified primary (FM55P and A375) and metastatic (FM55M2 and SK-MEL28) malignant melanoma cell lines, which were purchased from the European Collection of Authenticated Cell Cultures (FM55P and FM55M2), and the American Type Culture Collection (A375 and SK-MEL28), respectively. Experimental conditions on growth of the in vitro culture were described previously [5,16].

### 4.2. Drugs

Esculetin (ESC), mitoxantrone, docetaxel, paclitaxel and vemurafenib (Sigma-Aldrich, St. Louis, MO, USA) were dissolved in DMSO. Cisplatin (CDDP) and epirubicin (both from Sigma-Aldrich) were dissolved in phosphate buffered saline (PBS–CDDP) or in sterile hot water (epirubicin), respectively. To avoid any toxic effects of DMSO on cell proliferation, the concentrations of DMSO were maintained up to 0.1%.

### 4.3. Cell Viability Assessment

To determine the cell viability in all the studied malignant melanoma cell lines (FM55P, A375, FM55M2 and SK-MEL28) (density 2–3 × 10^4^ cells/mL depending on the cell line), normal human keratinocytes (HaCaT) (density 1 × 10^4^ cells/mL) and normal human melanocytes (HEMa-LP) (density 5 × 10^3^ cells/mL), the respective cells were placed on 96-well plates (NEST Biotechnology, Jiangsu, China), as described in detail earlier [16,17]. After 24 h, the culture medium was removed and increasing concentrations of esculetin and six commonly used chemotherapeutics (cisplatin, epirubicin, docetaxel, paclitaxel, mitoxantrone and vemurafenib) were added to the respective cells, as described earlier [16,17]. Each treatment protocol in the MTT assay was performed in triplicate and each experiment was repeated three times.

### 4.4. Cell Proliferation—BrdU Assay

To confirm cell proliferation in all the studied malignant melanoma cell lines (FM55P, A375, FM55M2 and SK-MEL28) after esculetin application, the ELISA BrdU Kit (Roche Diagnostics, Mannheim, Germany) was used. In this test, the optimized amounts of cells were placed on a 96-well plate (NEST) (100 µL/well). After 24 h, the cells were exposed to increased concentrations of esculetin for 48 h [6,17].

### 4.5. Cell Cytotoxicity—LDH Assay

To confirm cell cytotoxicity in all the studied malignant melanoma cell lines (FM55P, A375, FM55M2 and SK-MEL28), after esculetin application, Cytotoxicity Detection KitPLUS LDH (Roche Diagnostics, Mannheim, Germany) was used. In this test, the optimized amounts of all the studied malignant melanoma cells, normal human keratinocytes (HaCaT) and normal human melanocytes (HEMa-LP) were placed on 96-well plates (NEST). After 24 h, cells were washed in PBS, and increasing concentrations of esculetin were added to the cells. The cytotoxicity of esculetin was assessed by measuring cytoplasmic lactate dehydrogenase (LDH) activity released from damaged cells after 72 h of exposure to the former drug [16,17].

### 4.6. Isobolographic Analysis

To precisely assess pharmacodynamic interactions between esculetin and six chemotherapeutics in the malignant melanoma cell lines (FM55P, A375, FM55M2 and SK-MEL28) we used the isobolographic analysis in the MTT assay. Based on log-probit analysis according to Litchfield and Wilcoxon [54], we determined the percentage of inhibition of cell viability along with the respective concentration of tested substances: esculetin, cisplatin, epirubicin, docetaxel, paclitaxel, mitoxantrone and vemurafenib. In such a situation, the drug concentrations were transformed to logarithms and their anti-proliferative effects (measured by the MTT test) were transformed to probits. Both linearly related concentrations and anti-proliferative effects were plotted graphically into the Cartesian system of coordinates. The log-probit concentration–response lines allowed for calculating IC_50_ values for every tested drug. Parallelism of two concentration–response lines (i.e., between esculetin and cisplatin, epirubicin, docetaxel, paclitaxel, mitoxantrone and vemurafenib) was assessed with the test for parallelism, as described previously [5,55,56,57]. If two concentration–response effect lines are parallel, the additivity on the isobologram is illustrated as a diagonal line connecting the IC_50_ values for the tested compounds when used alone. However, if the concentration–response effect lines are not parallel, the additivity on the isobologram is illustrated as an area bounded by two (lower and upper) lines of additivity [58,59]. From the experimentally denoted IC_50_ values for the drugs administered alone, median additive inhibitory concentrations for the mixtures of esculetin + cisplatin, esculetin + epirubicin, esculetin + docetaxel, esculetin + paclitaxel, esculetin + mitoxantrone and esculetin +vemurafenib at the fixed ratio of 1:1 (IC_50add_) were calculated [17,58,59,60,61]. To determine the power (strength) of interactions, the interaction indices for all the studied combinations were calculated [48]. Owing to the interaction index (being a quotient of IC_50mix_ and IC_50add_ values), synergy can be observed if the interaction index is lower than the arbitrary accepted value of 0.7 [48,50,51]. Antagonism is recognized if the interaction index value for the studied combination is higher than the arbitrary accepted value of 1.3 [48,50,51]. Additivity is referred to the interaction index values ranged from 0.7 to 1.3 [48,50,51]. Finally, to illustrate the types of interactions between esculetin and all the tested chemotherapeutics in four malignant melanoma cell lines and briefly summarize the studied interactions, polygonogram-shaped graphs were depicted. On each graph, straight black lines indicated additive interactions, green lines illustrated synergistic interaction, whereas red lines depicted antagonism between drugs. The polygonogram shape is widely applied in isobolographic studies [62].

### 4.7. Statistical Analysis

Results were analyzed with a one-way ANOVA test followed by Tukey’s post hoc test. Values were presented as means ± standard errors (SEM). Statistical significance was marked graphically using asterisks (* *p* < 0.05, ** *p* < 0.01, *** *p* < 0.001, **** *p* < 0.0001, respectively). The inhibitory concentration (IC_50_ and the IC_50mix_) values for esculetin and six commonly used chemotherapeutics (cisplatin, epirubicin, docetaxel, paclitaxel, mitoxantrone and vemurafenib) were computed with the log-probit method [5,54,55]. The unpaired Student’s *t*-test with Welch’s correction was used to compare the experimentally derived IC_50mix_ values (for the two-drug mixtures of esculetin with cisplatin, epirubicin, docetaxel, paclitaxel, mitoxantrone or vemurafenib) with their respective, theoretically calculated and presumed to be additive IC_50add_ values, as recommended elsewhere [58,59].

## 5. Conclusions

Esculetin inhibited melanoma cell viability and proliferation in a dose-dependent manner. The IC_50_ for esculetin ranged from 18.20 ± 2.93 to 120.64 ± 30.39 μM depending on the malignant melanoma cell line. The combinations of esculetin with epirubicin and vemurafenib showed antagonistic interactions, the combinations with cisplatin, docetaxel and paclitaxel showed additive interactions. For the combination of esculetin with mitoxantrone, isobolographic analysis showed synergy. In the treatment of melanoma, esculetin should not be combined with epirubicin and vemurafenib due to the abolition of their effects, while the synergistic interaction (esculetin + mitoxantrone) deserves a preclinical recommendation as a very beneficial combination for anticancer therapies.

## Figures and Tables

**Figure 1 molecules-28-03889-f001:**
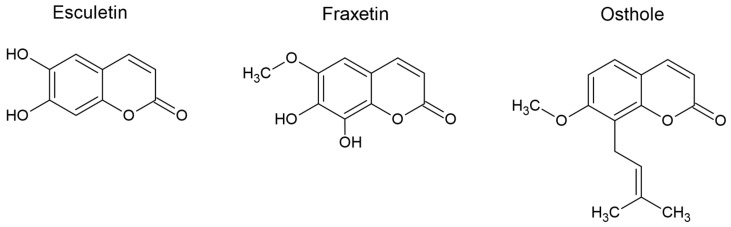
Structural formulas of naturally occurring simple coumarins (esculetin, fraxetin and osthole).

**Figure 2 molecules-28-03889-f002:**
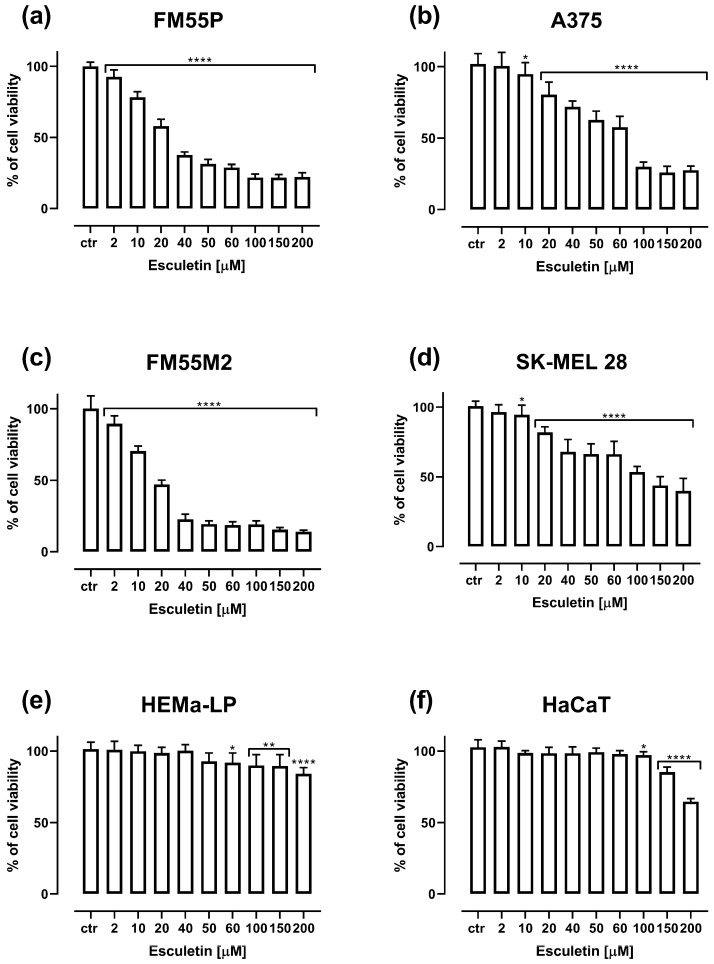
Influence of increasing concentrations of esculetin on the cell viability of malignant melanoma cell lines FM55P (**a**), A375 (**b**), FM55M2 (**c**), SK-MEL28 (**d**), normal human melanocytes HEMa-LP (**e**) and normal human keratinocytes HaCaT (**f**) in the MTT assay after 72 h. Columns represent means ± SEM. * *p* < 0.05, ** *p* < 0.01, and **** *p* < 0.0001 vs. the control (ctr) group.

**Figure 3 molecules-28-03889-f003:**
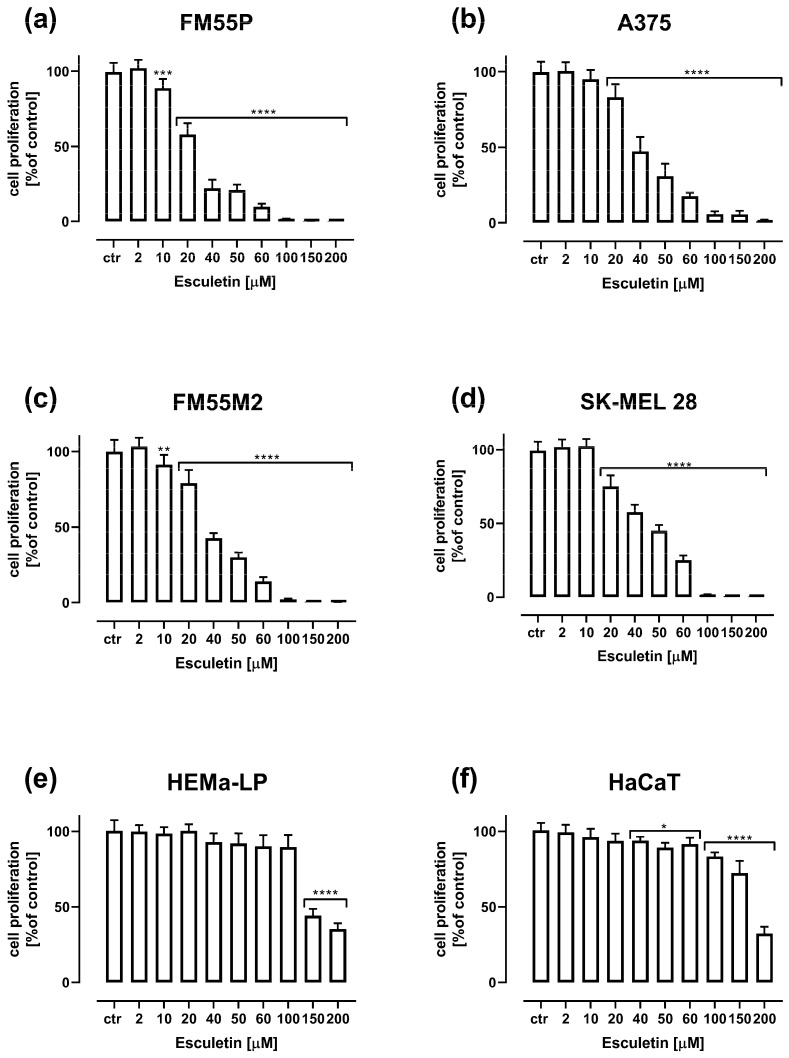
Influence of increasing concentrations of esculetin on the proliferation of malignant melanoma cell lines: FM55P (**a**), A375 (**b**), FM55M2 (**c**), SK-MEL28 (**d**), normal human melanocytes HEMa-LP (**e**) and normal human keratinocytes HaCaT (**f**) in the BrdU assay after 72 h. Columns represent means ± SEM. * *p* < 0.05, ** *p* < 0.01, *** *p* < 0.001 and **** *p* < 0.0001 vs. the control (ctr) group.

**Figure 4 molecules-28-03889-f004:**
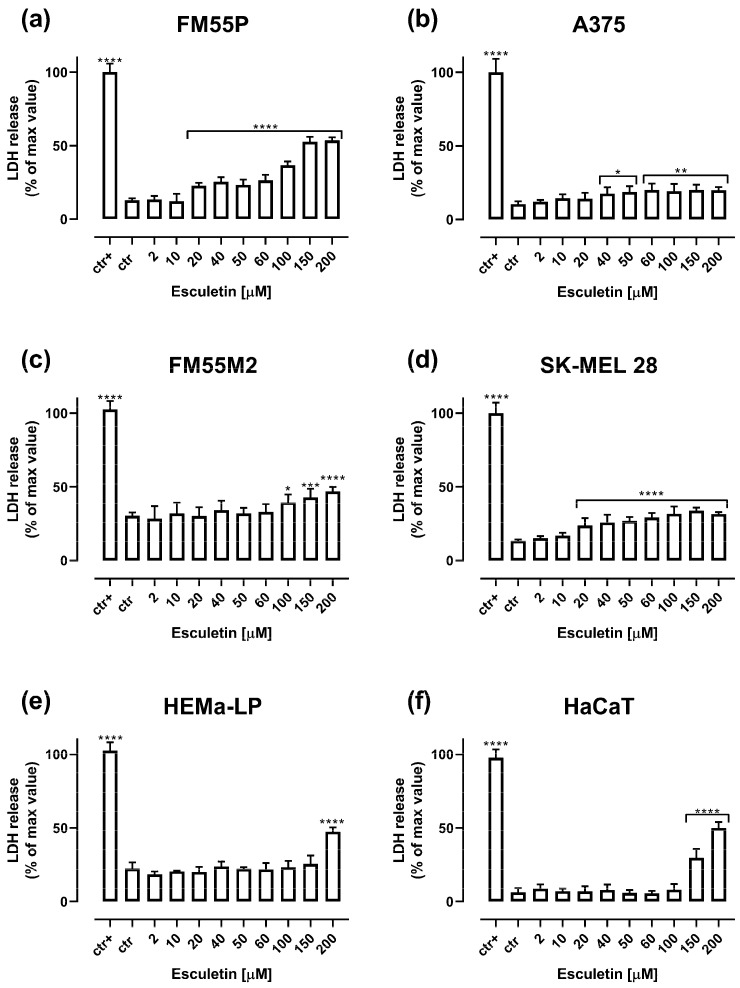
Influence of increasing concentrations of esculetin on cytotoxicity in malignant melanoma cell lines: FM55P (**a**), A375 (**b**), FM55M2 (**c**), SK-MEL28 (**d**), normal human melanocytes HEMa-LP (**e**) and normal human keratinocytes HaCaT (**f**) in the LDH assay. Columns represent means ± SEM. * *p* < 0.05, ** *p* < 0.01, *** *p* < 0.001 and **** *p* < 0.0001 vs. the control (ctr) group; ctr+—cells treated with lysis buffer.

**Figure 5 molecules-28-03889-f005:**
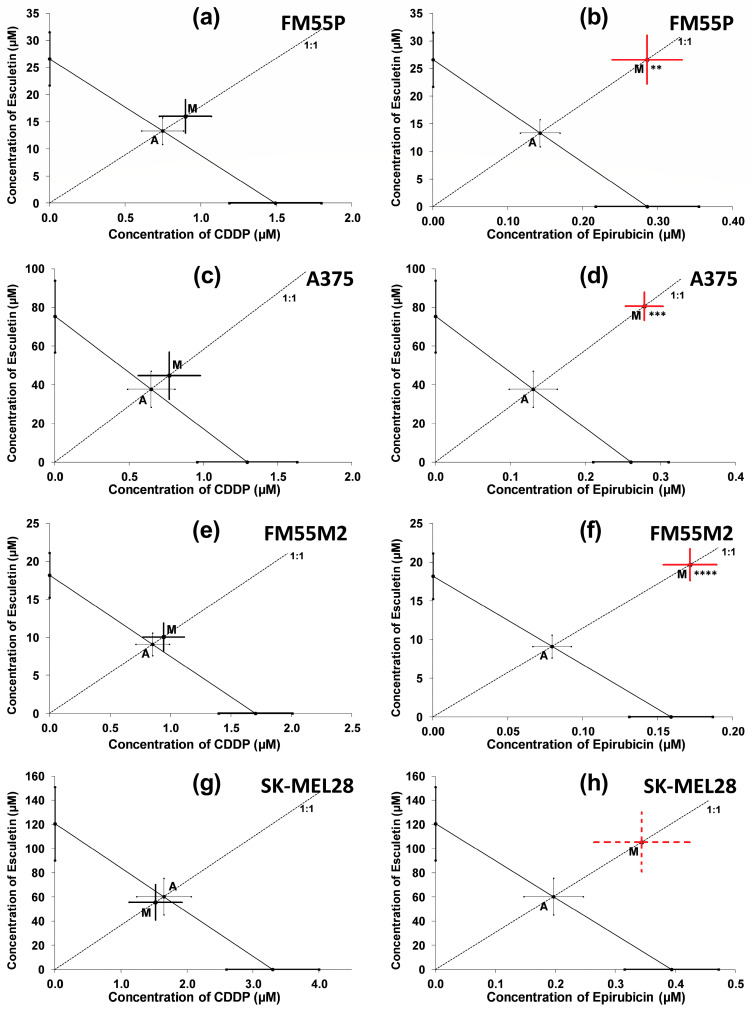
Interactions of esculetin with cisplatin (CDDP) (**a**,**c**,**e**,**g**), and esculetin with epirubicin (**b**,**d**,**f**,**h**) in the MTT assay in FM55P (**a**,**b**), A375 (**c**,**d**) FM55M2 (**e**,**f**), and SK-MEL28 (**g**,**h**) malignant melanoma cell lines. The IC_50_ (±SEM) values for esculetin, cisplatin (CDDP), and epirubicin are placed in the Cartesian system of coordination. The theoretically additive IC_50add_ values and the experimentally derived IC_50mix_ value are placed graphically as points A and M, respectively. Red lines indicate antagonistic interactions; dotted lines indicate tendency towards antagonism; ** *p* < 0.01, *** *p* < 0.001 and **** *p* < 0.0001 vs. the respective IC_50add_ value (Student’s *t*-test with Welch’s correction).

**Figure 6 molecules-28-03889-f006:**
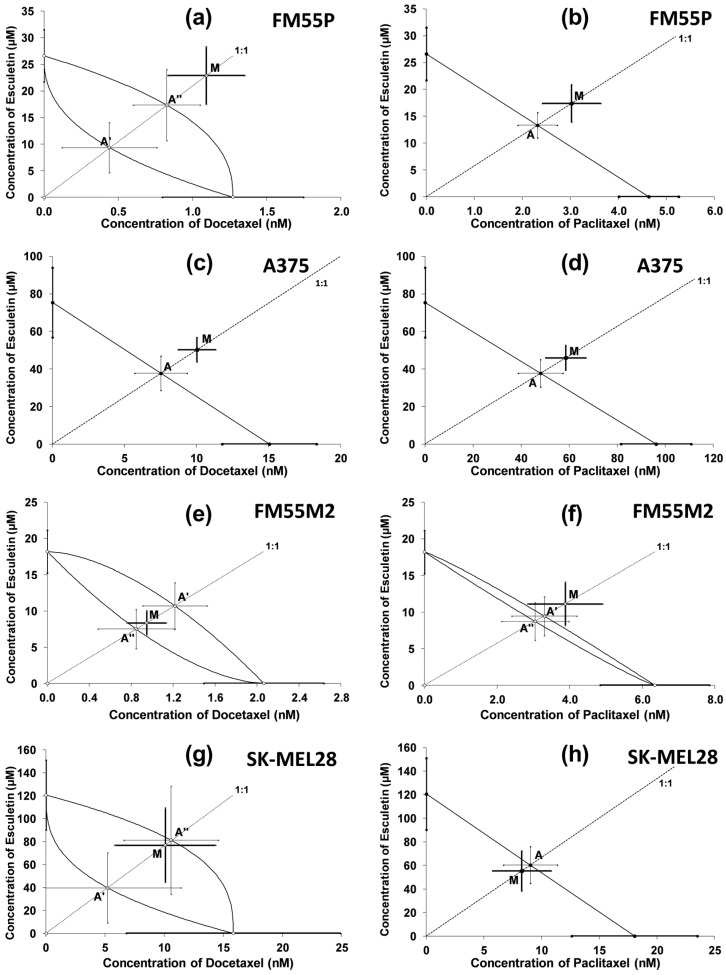
Isobolograms showing interactions between esculetin and docetaxel (**a**,**c**,**e**,**g**), and esculetin and paclitaxel (**b**,**d**,**f**,**h**) with respect to their anti-proliferative effects on FM55P (**a**,**b**), A375 (**c**,**d**) FM55M2 (**e**,**f**) and SK-MEL28 (**g**,**h**) malignant melanoma cell lines measured in vitro by the MTT assay. The IC_50_ ± SEM for ESC and docetaxel or paclitaxel are plotted on the Y- and X-axes, respectively. The points A, A′, A″ depict the theoretically calculated IC_50add_ values (±SEM) for both lower and upper isoboles of additivity. The point M on each graph represents the experimentally derived IC_50mix_ value (±SEM) for the mixture, which produced a 50% anti-proliferative effect (50% isobole) in the malignant melanoma cell lines. Black lines indicate additive interactions.

**Figure 7 molecules-28-03889-f007:**
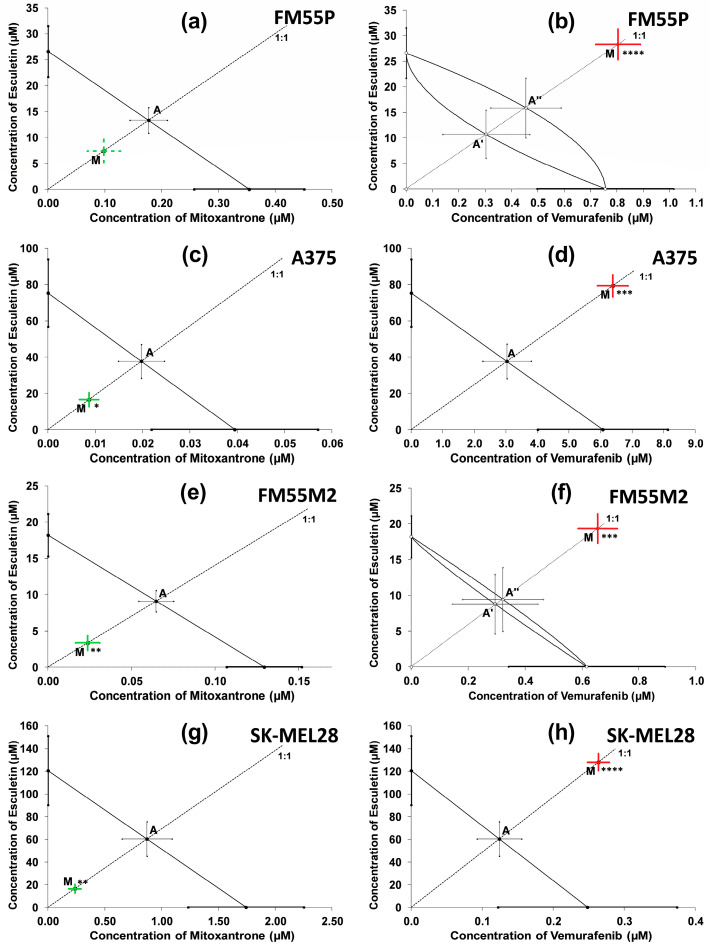
Isobolograms showing interactions between esculetin and mitoxantrone (**a**,**c**,**e**,**g**), and esculetin and vemurafenib (**b**,**d**,**f**,**h**) with respect to their anti-proliferative effects on FM55P (**a**,**b**), A375 (**c**,**d**), FM55M2 (**e**,**f**) and SK-MEL28 (**g**,**h**) malignant melanoma cell lines measured in vitro by the MTT assay. The IC_50_ ± SEM for ESC and mitoxantrone or vemurafenib are plotted on the Y- and X-axes, respectively. The points A, A′, A″ depict the theoretically calculated IC_50add_ values (±SEM) for both lower and upper isoboles of additivity. The point M on each graph represents the experimentally derived IC_50mix_ value (±SEM) for the mixture, which produced a 50% anti-proliferative effect (50% isobole) in the malignant melanoma cell lines. Red lines indicate antagonistic interactions; green lines indicate synergistic interactions; dotted lines indicate tendency towards synergy; * *p* < 0.05, ** *p* < 0.01, *** *p* < 0.001 and **** *p* < 0.0001 (Student’s *t*-test with Welch’s correction).

**Figure 8 molecules-28-03889-f008:**
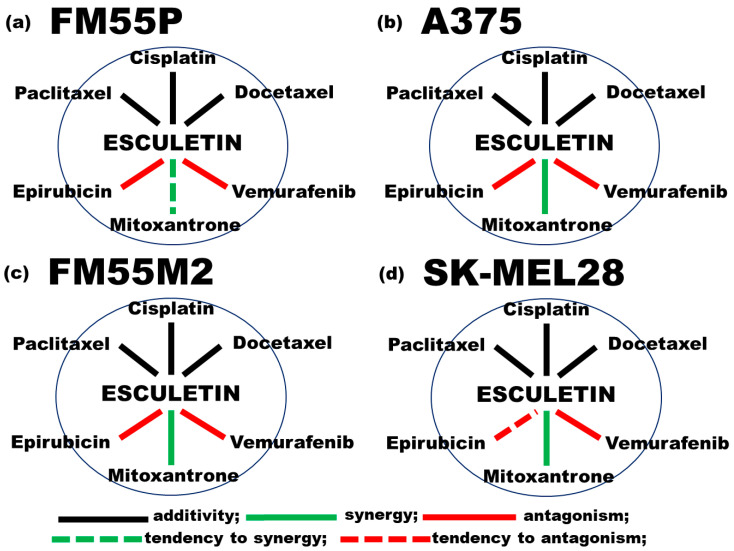
Graph in the shape of a polygonogram illustrates the interactions for two-drug mixtures among esculetin and six various cytostatic drugs in in vitro tests for four malignant melanoma cell lines.

**Table 1 molecules-28-03889-t001:** The anti-proliferative effects of esculetin, cisplatin, epirubicin, docetaxel, paclitaxel, mitoxantrone and vemurafenib in four various malignant melanoma cell lines detected in vitro by the MTT assay. The IC_50_ values are presented as means ± SEM.

Drug/Cell Line	FM55P	A375	FM55M2	SK-MEL28	References
Esculetin	26.62 ± 4.91 μM	75.38 ± 18.56 μM	18.20 ± 2.93 μM	120.64 ± 30.39 μM	this study
Cisplatin	1.49 ± 0.30 μM	1.29 ± 0.34 μM	1.70 ± 0.35 μM	3.30 ± 0.70 μM	[5,17]
Epirubicin	0.29 ± 0.07 μM	0.26 ± 0.05 μM	0.16 ± 0.03 μM	0.39 ± 0.08 μM	this study
Docetaxel	1.27 ± 0.55 nM	15.05 ± 3.27 nM	2.06 ± 0.66 nM	15.83 ± 9.05 nM	[16]
Paclitaxel	4.63 ± 0.62 nM	96.20 ± 14.61 nM	6.35 ± 1.74 nM	18.06 ± 6.29 nM	[16]
Mitoxantrone	0.35 ± 0.10 μM	0.04 ± 0.02 μM	0.13 ± 0.02 μM	1.74 ± 0.51 μM	[17]
Vemurafenib	0.76 ± 0.26 μM	6.07 ± 2.06 μM	0.62 ± 0.27 μM	0.25 ± 0.13 μM	this study

**Table 2 molecules-28-03889-t002:** Interaction index values for the combinations of esculetin with epirubicin, vemurafenib, mitoxantrone, cisplatin, paclitaxel and docetaxel in four various malignant melanoma cell lines detected in vitro by the MTT assay.

Drug Combination	Cell Line	IC_50add_	IC_50mix_	Interaction Index	Interaction
Esculetin + Epirubicin	A375	37.82	80.88 ***	2.14	Antagonism
	FM55M2	9.18	19.84 ****	2.16	Antagonism
	FM55P	13.45	26.90 **	2.00	Antagonism
	SK-MEL 28	60.52	105.76	1.75	Tendency to Antagonism
Esculetin + Vemurafenib	A375	40.72	85.76 ***	2.11	Antagonism
	FM55M2	9.41	20.00 ***	2.12	Antagonism
	FM55P	13.69	29.10 ****	2.13	Antagonism
	SK-MEL 28	60.44	128.34 ****	2.12	Antagonism
Esculetin + Mitoxantrone	A375	37.71	16.57 *	0.44	Synergy
	FM55M2	9.16	3.36 **	0.37	Synergy
	FM55P	13.49	7.47	0.55	Tendency to Synergy
	SK-MEL 28	61.19	16.61 **	0.27	Synergy
Esculetin + Cisplatin	A375	38.34	45.55	1.19	Additivity
	FM55M2	9.95	11.01	1.11	Additivity
	FM55P	14.06	16.91	1.20	Additivity
	SK-MEL 28	61.97	57.15	0.92	Additivity
Esculetin + Docetaxel	A375	45.22	60.26	1.33	Additivity
	FM55M2	8.36	9.30	1.11	Additivity
	FM55P	18.17	23.98	1.32	Additivity
	SK-MEL 28	91.77	86.98	0.95	Additivity
Esculetin + Paclitaxel	A375	85.79	104.51	1.22	Additivity
	FM55M2	12.76	15.01	1.18	Additivity
	FM55P	15.62	20.41	1.31	Additivity
	SK-MEL 28	69.35	63.58	0.92	Additivity

Interaction index is calculated as a quotient of the respective IC_50mix_ and IC_50add_ values; * *p* < 0.05, ** *p* < 0.01, *** *p* < 0.001 and **** *p* < 0.0001 (Student’s *t*-test with Welch’s correction).

## Data Availability

Data are contained within the article.

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
