# Peer review of "Synergy, Additivity and Antagonism between Esculetin and Six Commonly Used Chemotherapeutics in Various Malignant Melanoma Cell Lines—An Isobolographic Analysis"

_molecules, 2023, doi:10.3390/molecules28093889_

Round 1

Reviewer 1 Report

In the present manuscript, Paula et. al. investigated the drug combination effect of the herbal medicine Esculetin in different human malignant melanoma cell lines in combination with other well-known anti-cancer drugs. This combination therapy is useful in the context of effective clinical outcomes. The authors have recently published a similar study with another coumarin analogue fraxetin (Int. J. Mol. Sci. 2023, 24, 212) and another paper on anti-Parkinson drug Amantadine in Int J Mol Sci. 2022 Jul; 23(14): 7653. The current work builds on prior papers from the same group.

The study could be more significant over the previously published similar pattern of results from the same group by addressing the following points

(i)                  The authors should check the in vivo efficacy of this combination therapy to present an extended view.

(ii)                 Mutant cell lines, e.g., NRAS mutant-A375 can give an extended application of the protocol used in this study.

(iii)               The discussion section is unnecessarily long with the description of esculetin from other papers rather than describing the outcomes of this experiment. The discussion part should be modified focusing on the present experimental outcomes.

Author Response

The study could be more significant over the previously published similar pattern of results from the same group by addressing the following points

(i)    The authors should check the in vivo efficacy of this combination therapy to present an extended view.

Reply:

Thank you for your valuable suggestion. We are planning to perform in near future several experiments in a model of melanoma in mice to confirm the efficacy of selected coumarins in this in vivo model. But, in vitro experiments with various malignant melanoma cell lines (as a first screen test) provide us with information which of the tested combinations are worthy of being investigated in an in vivo study.

(ii)                 Mutant cell lines, e.g., NRAS mutant-A375 can give an extended application of the protocol used in this study.

Reply:

Thank you for your valuable suggestion. Definitely, the anticancer effects exerted by combinations of the studied drugs in BRAF mutant, NRAS mutant, KRAS mutant, and wild-type malignant melanoma cell lines should shed more light on molecular processes responsible for melanoma progression, proliferation and metastasizing.   

(iii)               The discussion section is unnecessarily long with the description of esculetin from other papers rather than describing the outcomes of this experiment. The discussion part should be modified focusing on the present experimental outcomes.

Reply:

Thank you for your valuable suggestion. The description of the anticancer effects of esculetin from other papers allowed us to explain its various molecular mechanisms involved in the anticancer effects. Additionally, following the other Reviewers’ suggestions the Discussion has been focused on the isobolographic studies and molecular aspects of interactions between esculetin and chemotherapeutics.

Reviewer 2 Report

The aim of this study was to evaluate the anticancer effects of esculetin and to assess pharmacodynamic interactions between esculetin and six commonly used cytostatics (cisplatin, epirubicin, docetaxel, paclitaxel, mitoxantrone and vemurafenib) using an isobolographic analysis. 

The authors have successfully identofied the combinations of esculetin with the existing drugs using different assays such as BrdU, LDH and finally isobolographic analysis.

But there are few problems that needs to be addressed-

1. There are moderate English corrections required. I would request for a proper check or analysis of the article with respect to English standards. There are a few typos and errors that also needs to be addressed. Further in the isobolographic section, there are contents written in red that indiactes the copy is a rough draft. So authors need to check the manuscript before sending it for the review.

2. How far isobolographic analysis is reliable in terms of finding the synergistic effect? That explanation needs to be given clearly

3. The authors can prove this conclusion of synergism between esculetin and Mitoxantrone using cytotoxicity assays.

4. The authors have mentioned about the molecular mechanism of apoptosis induction of cell lines using esculetin, but there is no evidence of western blotting or flow cytometry to prove the point. The authors must also prove their point of synergism between the drug and esculetin using some technique such as flow cytometry. 

The quality of English needs to be checked and there is a need of English correction

Author Response

  1. There are moderate English corrections required. I would request for a proper check or analysis of the article with respect to English standards. There are a few typos and errors that also needs to be addressed. Further in the isobolographic section, there are contents written in red that indiactes the copy is a rough draft. So authors need to check the manuscript before sending it for the review.

Reply:

The paper has been check for English grammar and style and corrected. The text marked in red was our explanations to the Editors’ suggestions and comments and it was not a rough draft.

  1. How far isobolographic analysis is reliable in terms of finding the synergistic effect? That explanation needs to be given clearly

Reply:

Following this suggestion we have added a new Table 2 illustrating the calculation of interaction index for each of the tested combinations. Due to the interaction index we can readily ascertain which of the investigated interactions were more synergistic or more antagonistic. A part of discussion about the notion of interaction index in isobolographic studies has been added.

  1. The authors can prove this conclusion of synergism between esculetin and Mitoxantrone using cytotoxicity assays.

Reply:

Thank you for your valuable suggestion. This study can be considered as a screen test allowing the selection of favorable interactions, which will be verified in further experiments using cytotoxicity assays.

  1. The authors have mentioned about the molecular mechanism of apoptosis induction of cell lines using esculetin, but there is no evidence of western blotting or flow cytometry to prove the point. The authors must also prove their point of synergism between the drug and esculetin using some technique such as flow cytometry.

 Reply:

Thank you for your valuable suggestion. This study can be considered as a screen test allowing the selection of favorable interactions, which will be verified in further experiments using western blotting and flow cytometry.

Reviewer 3 Report

This manuscript evaluated the anticancer effects of esculetin (a simple coumarin) and to assess pharmacodynamic interactions between esculetin and six commonly used cytostatics (cisplatin, epirubicin, docetaxel, paclitaxel, mitoxantrone and vemurafenib) on four human malignant melanoma cell lines (FM55P, A375, FM55M2 and SK-MEL28), by using an isobolographic analysis. The results demonstrated that esculetin inhibited, in a dose-dependent manner, malignant melanoma cell viability and proliferation. In the combinations of esculetin with mitoxantrone, the isobolographic analysis displayed synergy. In contrast, the combinations of esculetin with epirubicin and vemurafenib showed antagonistic interactions, and the combinations of esculetin with cisplatin, docetaxel, and paclitaxel showed additive interactions.

The manuscript provided a vital reference for the clinical combined use of esculetin. And that I would like to address the points as follows:

Minor comments:

1.In the clinic, esculetin and other commonly used cytostatics (cisplatin, epirubicin, docetaxel, paclitaxel, mitoxantrone, and vemurafenib) are different. Therefore, the synergy, additivity, and antagonism of their combination in various radio can not be generalized with only one ratio (like 1:1). Based on that, and it is suggested that the compatibility between esculetin and other therapy drugs in different ratios should be studied further.

2.The results have revealed that combinations of esculetin with mitoxantrone displayed a synergy effect. The combinations of esculetin with other drugs showed additive or antagonistic interactions.

I suggested the synergy, additivity, and antagonism mechanism of these drug compatibility be discussed from the point of view of the action target.

Minor editing of English language required。

Author Response

Minor comments:

1.In the clinic, esculetin and other commonly used cytostatics (cisplatin, epirubicin, docetaxel, paclitaxel, mitoxantrone, and vemurafenib) are different. Therefore, the synergy, additivity, and antagonism of their combination in various radio cannot be generalized with only one ratio (like 1:1). Based on that, and it is suggested that the compatibility between esculetin and other therapy drugs in different ratios should be studied further.

Reply:

Thank you for your valuable suggestion. In the Discussion we have mentioned the difference between proportions of drugs in combinations in clinical practice (based on the mass of the drugs) and isobolographic studies (based on the fractions of effects of the drugs). We agree with the Reviewer’s suggestion that one combination may exert synergy, additivity and even antagonism depending on the ratios tested, but in this study we used preferentially a fixed-ratio of 1:1, in which both drugs were used in equi-effective concentrations, being the fractions of the IC50 values of the studied drugs. A problem of selection of the adequate fixed-ratio combinations to preclinical verification in the MTT assay has been discussed. Besides, this in vitro study can be considered as a first screening, to select the combinations producing beneficial interactions. Such favorable interactions can be verified in further in vivo experiments.

2.The results have revealed that combinations of esculetin with mitoxantrone displayed a synergy effect. The combinations of esculetin with other drugs showed additive or antagonistic interactions.

I suggested the synergy, additivity, and antagonism mechanism of these drug compatibility be discussed from the point of view of the action target.

Reply:

We have discussed the observed synergistic and antagonistic interactions in relation to the molecular mechanisms of action of the studied drugs, as recommended.

Reviewer 4 Report

I have made some comments directly in the attached PDF.

Author Response

Reply:

All suggestions and comments presented in the PDF have been incorporated into the text of the revised version of the manuscript.

Round 2

Reviewer 1 Report

The authors gave the explanation of the queries as asked. But study on mutant cell lines and in vivo experiment will always provide better foundation of the experiment that has been described in this paper.

Reviewer 3 Report

The quality and logic of the revised manuscript are enhanced.

The compatibility between esculetin and other drugs in different ratios was discussed in the revised manuscript, though not added new experimental data.

I still suggested that the ratio of 1:1 should at least be discussed and cite similar clinical application examples.